# Systemic Administration of the TLR7/8 Agonist Resiquimod (R848) to Mice Is Associated with Transient, In Vivo-Detectable Brain Swelling

**DOI:** 10.3390/biology11020274

**Published:** 2022-02-10

**Authors:** Natalie May Zahr, Qingyu Zhao, Ryan Goodcase, Adolf Pfefferbaum

**Affiliations:** 1Department of Psychiatry and Behavioral Sciences, Stanford University School of Medicine, 401 Quarry Rd., Stanford, CA 94305, USA; qingyuz@stanford.edu (Q.Z.); dolfp@stanford.edu (A.P.); 2Neuroscience Program, SRI International, 333 Ravenswood Ave., Menlo Park, CA 94025, USA; ryan.goodcase@gmail.com

**Keywords:** inflammation, brain, edema

## Abstract

**Simple Summary:**

Physiological changes to the body can affect the brain. For example, inflammation that originates in the body can be sensed by the brain. In this study, we used an agent called resiquimod (R848) to stimulate inflammation in the periphery and measured structural and metabolic brain responses in anesthetized mice using in vivo magnetic resonance imaging. Relative to baseline prior to drug administration, a high dose of R848 caused sickness behaviors and volume expansion in several cortical regions at 3 h that were no longer evident at 24 h. Transient volume expansion in response to peripheral immune stimulation is consistent with brain swelling.

**Abstract:**

Peripheral administration of the *E. coli* endotoxin lipopolysaccharide (LPS) to rats promotes secretion of pro-inflammatory cytokines and in previous studies was associated with transient enlargement of cortical volumes. Here, resiquimod (R848) was administered to mice to stimulate peripheral immune activation, and the effects on brain volumes and neurometabolites determined. After baseline scans, 24 male, wild-type C57BL mice were triaged into three groups including R848 at low (50 μg) and high (100 μg) doses and saline controls. Animals were scanned again at 3 h and 24 h following treatment. Sickness indices of elevated temperature and body weight loss were observed in all R848 animals. Animals that received 50 μg R848 exhibited decreases in hippocampal N-acetylaspartate and phosphocreatine at the 3 h time point that returned to baseline levels at 24 h. Animals that received the 100 μg R848 dose demonstrated transient, localized, volume expansion (~5%) detectable at 3 h in motor, somatosensory, and olfactory cortices; and pons. A metabolic response evident at the lower dose and a volumetric change at the higher dose suggests a temporal evolution of the effect wherein the neurochemical change is demonstrable earlier than neurostructural change. Transient volume expansion in response to peripheral immune stimulation corresponds with previous results and is consistent with brain swelling that may reflect CNS edema.

## 1. Introduction

Immune responses to potential pathogens are generally determined by engagement of specific pattern recognition receptors such as Toll-like, NOD-like, RIG-I-like, and C-type lectin receptors [1]. Toll-like receptors (TLRs) play a critical role in innate immunity [2]. Activation of TLR7/8 receptors is associated with cytotoxic T-cell responses, concomitant blocking of immunosuppressive cells [3,4,5,6,7], and upregulation of genes involved in inflammation such as pro-inflammatory cytokines (TNFα, IL6) [8] and type I interferons (IFN) [9,10,11,12]. Older literature suggests that TLR7 is present in a variety of CNS cell types, including neurons [13,14]. Recent reports, however, suggest preferential TLR7 distribution to microglia [15,16].

A number of ligands have been designed to boost (agonists) or block (antagonists) inherent signal transduction at TLR7/8 receptors. They are of interest in acute applications, such as adjuvants to vaccines [17,18], and in chronic protocols, such as immunotherapies for cancer [19,20]. There is also emerging evidence that TLR7/8 receptors may be an appropriate target for the treatment of alcohol-use disorders (AUD) e.g., [21,22,23]. One TLR7/8 agonist, imiquimod, has FDA approval [24]. In preclinical studies, topical imiquimod (i.e., Aldara cream) was associated with systemic pro-inflammatory cytokine production, enlargement of spleen and draining lymph nodes [25], induction of CNS chemokine and interferon signaling, infiltration of immune cells into the CNS [26,27], and increased Iba-1 and GFAP staining indicating micro- and astro- gliosis [28]. Imiquimod given at 50 μg intraperitoneal (i.p.) increased depressive symptoms in mice 2 h after treatment, possibly via elevated IFNα expression [29].

R848 (Resiquimod) is a synthetic, imidazoquinoline (i.e., tricyclic organic) compound able to activate immune cells via TLR7/8 MyD88-dependent signaling [30,31] with a much higher potency than imiquimod [32], but may have species (i.e., rodent vs. non-rodent) specific activity [33]. In rodents, administration of R848 (i.p.) induces acute sickness responses including hypophagia, body weight loss, and decreased voluntary locomotor activity [15,34,35]. In longitudinal evaluation, CNS pro-inflammatory gene expression (e.g., TNFα and IL6 peak at 4–6 h) and glial morphology alterations (e.g., increased Iba1 signal intensity) in response to R848 exposure were shown to undergo rapid tachyphylaxis [15].

A single injection of the *E. coli* endotoxin lipopolysaccharide (LPS, i.p.), which targets TLR4 [36,37], similarly induces acute (i.e., 2 h) pro-inflammatory cytokine signaling [38,39] and sickness responses [40]. In response to acute LPS-induced peripheral inflammation in rats, in vivo magnetic resonance imaging (MRI) and spectroscopy (MRS) revealed transient expansion of posterior midline CNS tissue volumes [41]— which may represent acute brain edema e.g., [42,43]—and transient elevations in striatal glutamine [41]. The current study, therefore, tested the hypothesis that peripheral TLR7/8 stimulation of mice with R848 would similarly result in MR-detectable changes to the brain including modulation of the glutamate/glutamine balance and brain swelling consistent with edema. To the best of our knowledge, only a single study has previously evaluated the in vivo effects of R848 using MRI but focused on cardiac, not brain, tissue [44].

## 2. Materials and Methods

### 2.1. Ethics Statement

All experimental procedures were conducted in accordance with the Guide for the Care and Use of Laboratory Animals of the National Institutes of Health. The Institutional Animal Care and Use Committees at SRI International and Stanford University approved all procedures.

### 2.2. Animals and Drugs

A total of 24 male C57Bl mice (Jackson Laboratories, Sacramento, CA, USA), arriving at 6 weeks of age (24.88 ± 1.7 g), were used in this study. Only male mice were included to minimize sex effects [45,46,47] and to expand on previous work [15]. Mice were housed three per cage, maintained in a pathogen-free facility on a 12-h light/dark cycle, and had ad libitum access to regular chow and water.

Resiquimod (R848, Invivogen, San Diego, CA, USA; catalog #tlrl-R848-5, version #16F20-MM) was dissolved in endotoxin-free water to 1 mg/mL and injected intraperitoneally (i.p.) at doses of 50 μg (~2 mg/kg) or 100 μg (~4 mg/kg) in experimental (R848) mice, while control animals received saline (0.9% NaCl, 100 μL). These R848 doses were chosen based on previous work in mice showing a detectable response at 10 μg and a more robust immune response at 100 μg [15,21,48]. Single use, sterile needles (27.5 gauge) were used to administer treatments.

### 2.3. Magnetic Resonance (MR) Scanning Procedures and Data Analysis

Each mouse was scanned three times with one scan per day over three consecutive days. The initial scan on day 1 was a baseline (scan 1). On day 2, mice were injected with either saline or R848 and scanned 3 h later (scan 2). On day 3, each animal underwent a final scan (scan 3), 24 h after the saline or R848 injection. These scan time points—3 h and 24 h after injection—were chosen because the literature reports optimal stimulation of the immune system (e.g., IFN-α and IFN-γ secretion) at 3 h and resolution typically by 24 h [49,50]. For each scan, mice were anesthetized with isoflurane (3% for induction; ~0.5–2% for maintenance during the scan); body weight and temperature were acquired and recorded; then animals were placed on the cradle base with built-in water circulation for body temperature control. Temperature and respiration were monitored throughout each scan (∼1 h). All animals received subcutaneous saline (0.5 mL) for hydration at the end of the scan.

#### 2.3.1. MR Spectroscopy

Point RESolved Spectroscopy (PRESS) data were obtained from two voxels [bilateral hippocampus (6.0 × 1.5 × 2.5 mm^3^) and right dorsal striatum (3.0 × 3.0 × 2.5 mm^3^) (Figure 1a)] with TR = 2500 ms, TE = 16 ms, 760 points, spectral width 13.3177 ppm, receiver bandwidth 4000 Hz, and with VAPOR (variable power and optimized relaxation delays) outer volume water suppression (bandwidth 200 Hz). Water suppressed excitations (256) were at 2.35 ppm (the frequency of glutamate) and an additional unsuppressed acquisition (32 excitations) was acquired at 4.7 ppm for water scaling.

Metabolite concentrations were quantified with LCModel with eddy-current correction and water scaling (which accounts for CSF volume in the voxel) using a basis set of 21 metabolites. The analysis was run with the “water reference” option that provided reasonably meaningful absolute metabolite concentrations based on the unsuppressed water content of the voxel. The analysis window was 0.2–4.0 ppm. Data were pre-processed with zero-order phasing, referencing, and residual water line removal. Data were fit to a linear combination of a number of metabolites in a simulated basis set designed for Bruker 7T MRS data acquisition at TE = 16 ms provided by Stephen Provencher [sp@lcmodel.CA] [51,52] containing alanine, creatine (Cr), phosphocreatine (PCr), glutamine (Gln), glutamate (Glu), glycerophosphorylcholine (GPC), phosphorylcholine (PCh), glutathione (GSH), inositol (Ins), lactate, N-acetylaspartate (NAA), N-acetylaspartylglutamate (NAAG), scyllo-inositol, taurine, and several lipids and macromolecules (Figure 1b). Only metabolite concentrations derived from fitted spectra consistently within average Cramér–Rao bounds <15% were considered (Table 1). 

#### 2.3.2. Structural Magnetic Resonance Imaging (MRI)

MR data were collected on a Bruker 70/16 US AVANCE III 7.0T system (Karlsruhe, Germany) with 380 mT/m gradient strength on each (X, Y, and Z) axis, slew rate of 3420 T/m/s, 16 cm bore size using a Bruker mouse head volume coil (23 mm) and ParaVision 6.1 software. A gradient-recalled echo (GRE) localizer scan was used to position the animals in the scanner and for graphical prescription of the subsequent scans. Structural MR data analysis was based on acquisition of T2-weighted, high-resolution, TurboRare sequences: repetition time (TR) = 6774.8 ms; echo time (TE) = 33 ms; field of view (FOV) = 18 × 18; matrix = 144 × 144; pixel size = 0.125 × 0.125 × 0.5 mm^3^; 4 averages; echo spacing = 11 ms; rare factor (i.e., echo train length) = 8; slice thickness= 0.5 mm; 40 slices.

Structural MRI data preprocessing of each image included removal of noise [53] and inhomogeneity correction via ANTS 2.1.0 [54]. Each image was skull stripped by aligning a template to the scan via symmetric diffeomorphic registration [55], and the resulting deformation map was applied to the brain mask of the template. Image inhomogeneity correction was repeated on skull-stripped images. The structural template was segmented into CSF, gray matter, and white matter by fitting the histogram of image intensities with three Gaussians yielding the probability of the three tissue types on a voxel-by-voxel basis. A CSF mask was constructed comprising all voxels in which the probability of CSF was the largest of the three values (i.e., CSF volume, Table 2). In parallel, the template was parcellated into 42 regions-of-interest (ROIs) by first registering the Allen Reference Atlas (ARA) [56] to it and then projecting the CSF mask onto the ARA parcellation map. Finally, tissue segmentation and ARA parcellation were transformed to the space of each image using the prior deformation map and resampled in the template space by rigidly aligning the bias-corrected skull-stripped image to the template. Volume of the three tissue types within each ROI was calculated by Computational Morphometry Toolkit (CMTK) [57].

Of the 42 ARA atlas-defined ROIs, one ROI (cerebellar nuclei) was removed from analysis as the measure was deemed unreliable. The ARA olfactory cortex was divided into olfactory bulb and olfactory cortex. The remaining 41 ROIs were reduced to 19 ROIs by adding individual volumes as follows: **fronto-orbital cortex** = frontal pole cerebral cortex + orbital area; **motor cortex** = primary motor area + secondary motor area; **somatosensory cortex** = primary somatosensory area trunk, lower limb, nose, upper limb, barrel field, mouth + supplemental somatosensory area; **insular cortex** = gustatory areas + visceral area + agranular insular area; **temporal cortex** = auditory areas + temporal association areas; **visual cortex** = visual areas + retrosplenial area + posterior parietal association areas; **cingulate cortex** = anterior cingulate area + infralimbic area + prelimbic area; **hippocampal formation** = ectorhinal area + perirhinal area + hippocampal region + retrohippocampal region; **striatum** = striatum dorsal region + striatum ventral region + lateral septal complex + striatum-like amygdalar nuclei; **thalamus** = thalamus sensory-motor cortex related + thalamus polymodal association cortex related. The remaining volumes (i.e., olfactory bulb, olfactory cortex, cortical subplate, pallidum, hypothalamus, midbrain, pons, medulla, cerebellar cortex) were treated individually (i.e., not combined).

### 2.4. Statistics

Statistical analyses were conducted using JMP^®^ Pro 16.0.0 (SAS Institute Inc., Cary, NC, USA, 1989–2021). Outlier values (i.e., >3 SD from mean) were winsorized as follows: for hippocampal PCr at scan 2 for a mouse treated with 100 μg R848 (from 2.62 to 7.09 I.U.); for volume of the medulla at scan 2 for a mouse treated with 50 μg R848 (from 24.84 to 31.29 mm^3^). Analyses included three-group (control, 50 μg R848, 100 μg R848), repeated-measures (baseline, 3 h, 24 h) multivariate analysis of variance (MANOVAs), followed by separate three-group ANOVAs per time point, and then two-group *t*-test comparisons. Non-parametric Spearman’s ρ correlational analysis was conducted where relevant.

## 3. Results

### 3.1. Sickness Responses

A three-group by three-time point MANOVA on body weight (expressed as percent of baseline) was significant (F_4,42_ = 11.9, *p* < 0.0001, Figure 2a). Body weight did not distinguish the three groups at scan (all normalized to zero) or scan 2 (F_2,24_ = 0.6, *p* = 0.57); the three-group ANOVA was significant at scan 3 (F_2,24_ = 20.2, *p* < 0.0001). At 24 h following injection, the 50 μg R848 group had lost 5.5% of their baseline body weight; the 100 μg R848 group lost 2.8% of their baseline body weight; the control group was unaffected.

A three-group by three-time point MANOVA on temperature was significant (F_4,40_ = 3.6, *p* = 0.01; Figure 2b; temperature data missing for one 50 μg R848-treated mouse). Temperature did not distinguish the three groups at baseline (F_2,22_ = 0.4, *p* = 0.66). At scan 2, 3 h following injection (F_2,22_ = 5.2, *p* = 0.02), the 50 μg R848-treated animals had a higher temperature than saline-treated mice (*p* = 0.004); the 100 μg R848 group had nominally higher temperature relative to the saline-treated group (*p* = 0.09); the 50 μg and 100 μg R848 treated animals were not different (*p* = 0.16). At scan 3, the three-group ANOVA was marginally significant (F_2,22_ = 3.1, *p* = 0.07): the 100 μg R848-treated mice had higher temperatures than the control animals (*p* = 0.02) but were not different from the 50 μg R848 treated animals (*p* = 0.24); temperature did not distinguish the 50 μg R848-treated and control mice at the third scan (*p* = 0.25).

### 3.2. MRS Metabolite Changes

Fifteen metabolites had average Cramér–Rao bounds <15% (Table 1). Two metabolites showed significant R848 treatment effects in the hippocampus, identified with three-group by three-time point MANOVAs: *N*-acetylaspartate (NAA, F_4,42_ = 4.7, *p* = 0.003) and phosphocreatine (PCr, F_4,42_ = 5.0, *p* = 0.002) (Figure 3). For both metabolites at scan 2, 3 h following R848 treatment, the 50 μg R848-treated group had lower NAA and lower PCr than the 100 μg R848-treated or saline groups. The overall pattern for the transient changes in hippocampal metabolite levels to R848 treatment (i.e., at the 3 h time point relative to baseline) indicated a minimal response for the saline group (NAA −0.03 ± 11.7%; PCr −1.3 ± 2.7%), a moderate decline in levels for the 50 μg R848-treated group (NAA −13.1 ± 17.2%; PCr −28.4 ± 16.2%), and a nominal increase for the 100 μg R848-treated group (NAA 7.0 ± 9.2%; PCr +20.9 ± 34.5%). In the striatum, three-group by three-time point MANOVAs were significant for glutamine (Gln, F_4,42_ = 3.83, *p* = 0.01) and phosphocholine (PCh, F_4,42_ = 2.60, *p* = 0.05) (Table 1). Follow-up ANOVAs, however, failed to identify group differences at any scan. Instead, the MANOVAs were significant due to changes in striatal metabolite levels in all three groups between scans 1 and 2: for Gln, a +10.0% increase in the 50 μg R848-treated group and decreases in the 100 μg R848-treated (−7.2%) and saline (−5.2%) groups; for PCh, a +16.3% increase in the saline group and decreases in the 50 μg (−8.1%) and 100 μg (−13.1%) R848-treated groups.

### 3.3. Structural MRI Volumetric Changes

Of the 20 ROIs listed in Table 2, only four were responsive to R848 treatment: motor (F_4,42_ = 2.9, *p* = 0.04), somatosensory (F_4,42_ = 3.2, *p* = 0.02), and olfactory (F_4,42_ = 3.7, *p* = 0.01) cortices; and pons (F_4,42_ = 3.0, *p* = 0.03) (Figure 4 and Figure 5). At scan 2 (3 h following injections), three-group ANOVAs were significant for motor (F_2,24_ = 4.7, *p* = 0.02), somatosensory (F_2,24_ = 3.6, *p* = 0.05), and olfactory (F_2,24_ = 3.7, *p* = 0.04) cortices; and pons (F_2,24_ = 3.7, *p* = 0.04). For all four ROIs, the 100 μg R848 group had larger volumes than the saline or 50 μg R848-treated groups. At scan 3 (24 h following injections), the ANOVAs no longer distinguished groups. The overall pattern for the transient ROI responses to R848 treatment (i.e., percent change at the 3 h relative to the baseline time point, averaged across the 4 ROIs) was the following: the saline group showed a nominal decline in volume (−1.7 ± 5.1%), the 50 μg R848-treated group showed a moderate decline in volume (−4.0 ± 3.8%), and the 100 μg R848-treated group showed an increase in volume (+5.10 ± 4.9%). The three-group by three-time point MANOVA for insular volume was significant (F_4,42_ = 4.4, *p* = 0.005), but was driven by the incidentally larger baseline volume of the insula in the to-be-treated 50 μg R848 group.

### 3.4. Correlations between Brain and Response Measures

Non-parametric Spearman ρ evaluated relationships among variables. In the 50 μg R848-treated group, relationships between changes in hippocampal metabolite levels, body weight, temperature, and ROI volumes were not forthcoming. For ROI volumes, only the 100 μg R848-treated group was considered. The percent change in body weight between scan 1 and scan 3 correlated with percent change in volume between scan 1 and scan 2 for somatosensory cortex (r = −0.79, *p* = 0.02) and olfactory cortex (r = −0.73, *p* = 0.04) (Figure 6).

## 4. Discussion

To the best of our knowledge, this is the first in vivo imaging study of the brain in response to peripheral R848 administration. The body weight loss observed at 24 h in the both 50 μg and 100 μg R848-treated mice is consistent with “sickness behavior” and previous observations, reported at doses as low as 10 μg [15,21]. Regarding temperature, previous studies suggest transient increases in temperature (e.g., at 2 h following R848) taken during the day, although there is some indication that body temperature in response to R848 may decrease at night [15,21].

Consistent with our previous study in rats in response to TLR4 activation with LPS, the current study in mice demonstrates transient volume expansion in cortical regions (including motor, somatosensory, and olfactory cortices) and pons. The time course of in vivo volume changes is consistent with the temporal profile of molecular changes, demonstrating acute inflammatory responses at early intervals, but virtually no residual responses [15,58]. Acknowledging that the regional distribution of TLR expression may depend on the type (e.g., hypoxia vs. concussion) and length of exposure to insult [59,60], the localization herein of the R848 response, to cortical regions, is compatible with evidence for TLR7 expression to neocortex of the developing mouse brain [61] and prominent cortical distribution in the adult mouse brain [16,62]. This pattern is in contrast to the LPS response in rats, which was localized to splenial, retrosplenial, and peri-callosal hippocampal regions, but comports with a predominately hippocampal distribution of TLR4 [63,64,65].

Transient metabolic changes observed at the 50 μg but not 100 μg dose may represent a continuous CNS response to the degree of peripheral inflammation. In other words, as there is evidence for a pro-inflammatory response to R848 to be concentration-dependent e.g., [66], lower concentrations of R848 with a lower pro-inflammatory response may only stimulate metabolic changes while higher doses associated with more pronounced pro-inflammation may result in brain swelling. Transient changes in the levels of NAA and PCr more likely reflect altered energy utilization than osmotic imbalance [67,68], which may manifest before volumetric changes evolve.

In summary, the current study recapitulates in vivo changes in body weight and temperature in response to peripherally administered R848 and expands on previous findings by demonstrating an in vivo CNS response to R848. The brief metabolic changes at the lower dose may reflect energy imbalance and the transient volumetric changes (i.e., swelling) at the higher dose may represent brain edema in response to peripheral immune stimulation. Remaining to be elucidated is the mechanism of pro-inflammatory signal transduction from the periphery to the brain which may include stimulation of the vagus nerve; activation of cerebral endothelial cells; or direct access to brain parenchyma via circumventricular organs.

## Figures and Tables

**Figure 1 biology-11-00274-f001:**
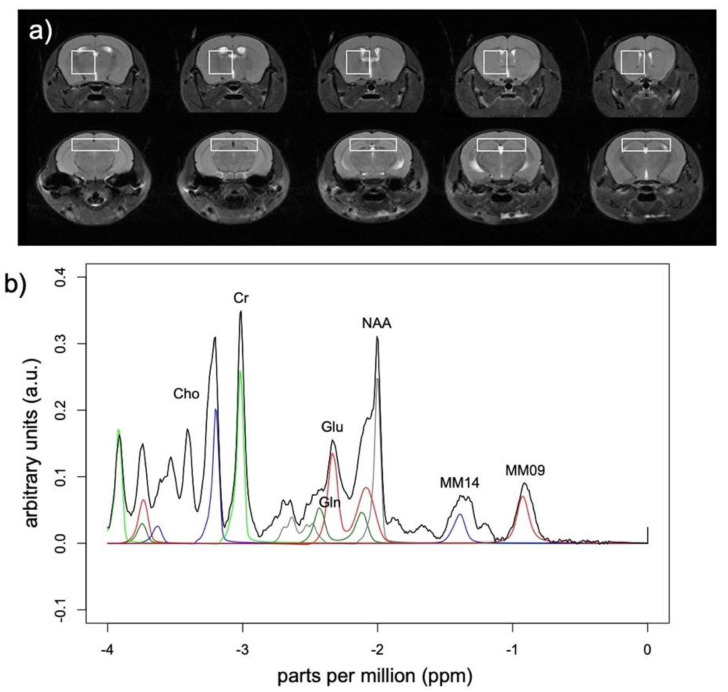
Magnetic resonance spectroscopy (MRS) voxel location and exemplary spectra. (**a**) Voxels localized to striatum (top) and hippocampus (bottom) presented on coronal slices from posterior to anterior. (**b**) Group average spectra from 8 animals at baseline before receiving 100 μg R848. LCModel fits for Cho = glycerophosphorylcholine + phosphorylcholine in blue, Cr = creatine + phosphocreatine in green, Glu = glutamate in red, Gln = glutamine in blue, NAA = *N*-acetylaspartate in gray, MM14 and MM09 = macromolecules at 1.4 and 0.9 ppm in blue and red respectively.

**Figure 2 biology-11-00274-f002:**
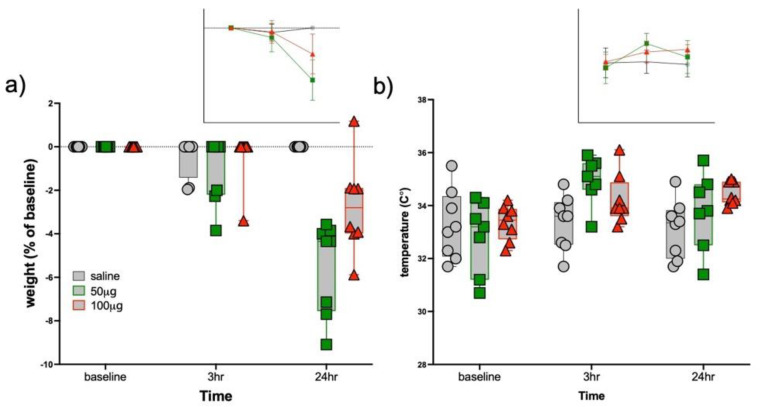
Evidence for sickness behaviors in R848-treated animals. (**a**) Body weight at each scan relative to baseline scan body weights. (**b**) Temperature taken before each scan. Control = gray, 50 μg R848 = green, 100 μg R848 = red.

**Figure 3 biology-11-00274-f003:**
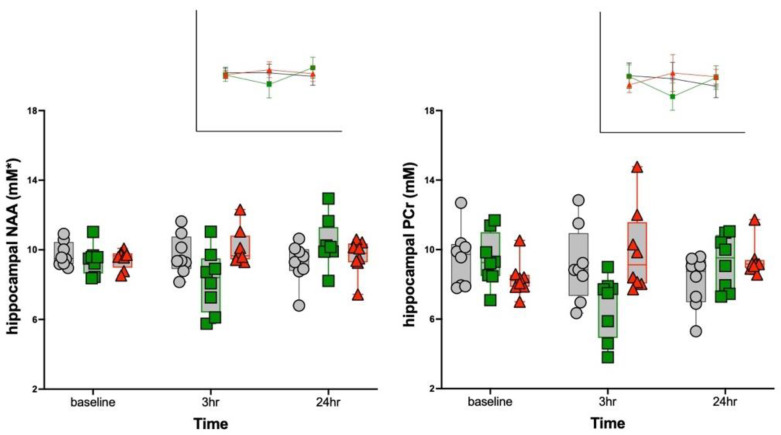
Hippocampal *N*-acetylaspartate (NAA) and phosphocreatine (PCr) levels of the three groups at the three time points. Insets display mean ± SD for each group at each time point. Control = gray, 50 μg R848 = green, 100 μg R848 = red. * mM/kg wet weight.

**Figure 4 biology-11-00274-f004:**
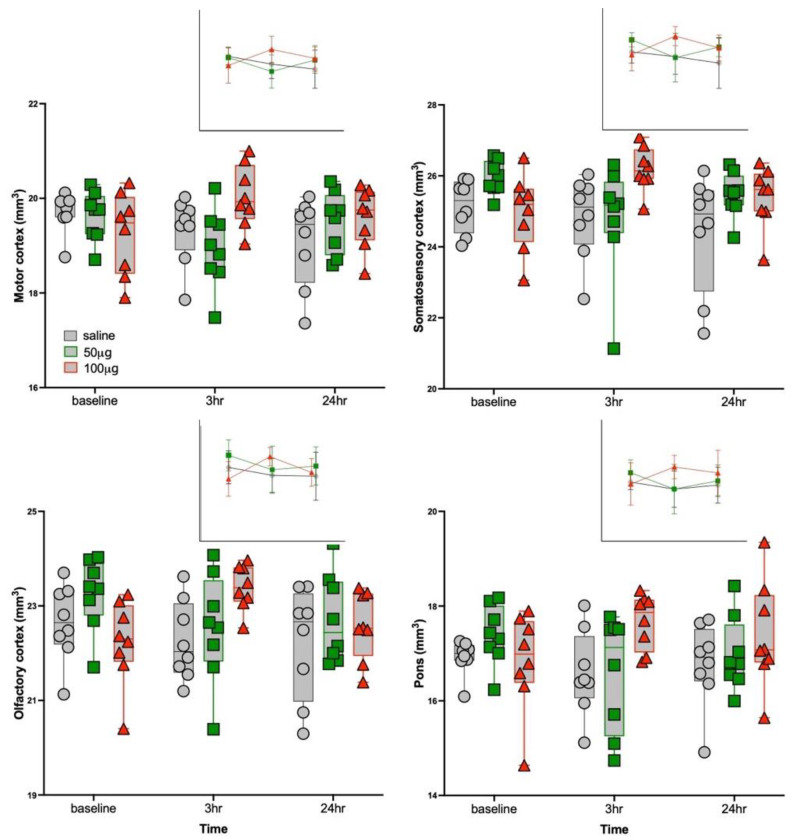
Regional brain volumes (motor, somatosensory, olfactory, and pons) of the three groups at the three time points. Insets display mean ± SD for each group at each time point. Control = gray, 50 μg R848 = green, 100 μg R848 = red.

**Figure 5 biology-11-00274-f005:**
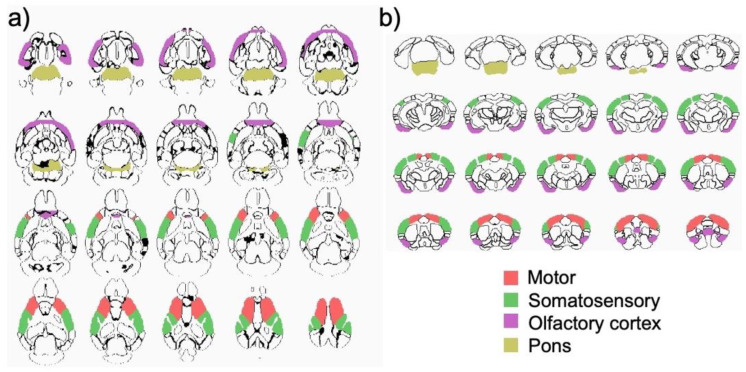
Color-coded regions of interest (ROIs) projected on an outline of the Allen Reference Atlas (ARA) parcellations in (**a**) axial and (**b**) coronal views. Motor cortex = red, Somatosensory cortex = green, Olfactory cortex = magenta, Pons = mustard.

**Figure 6 biology-11-00274-f006:**
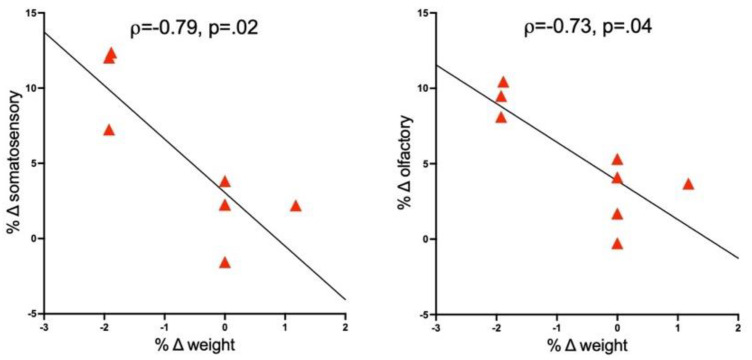
Correlations between changes in body weight and changes in regional brain volumes (i.e., somatosensory and olfactory cortices) between scans 1 and 2 in the 100 μg R848-treated group.

**Table 1 biology-11-00274-t001:** Statistical results for MRS metabolites with average Cramér-Rao bounds <15% in hippocampal and striatal voxels.

	Hippocampus					Striatum				
Metabolite	3-Group × 3-Time MANOVAs		3-Group ANOVAs			3-Group × 3-Time MANOVAs		3-Group ANOVAs		
			Baseline	3 h	24 h			Baseline	3 h	24 h
	F_(4,42)_ ^+^	*p*-Value	F_(2,24)_, *p*	F_(2,24)_, *p*	F_(2,24)_, *p*	F_(4,42)_ ^+^	*p*-Value	F_(2,24)_, *p*	F_(2,24)_, *p*	F_(2,24)_, *p*
Cr + PCr	1.61	0.19	n.s.	n.s.	n.s.	2.04	0.11	n.s.	n.s.	n.s.
GABA	1.68	0.17	n.s.	n.s.	n.s.	0.14	0.97	n.s.	n.s.	n.s.
Gln	0.24	0.91	n.s.	n.s.	n.s.	3.83	**0.01**	2.3, 0.12	2.2, 0.13	0.5, 0.60
Glu	0.61	0.66	n.s.	n.s.	n.s.	1.39	0.25	n.s.	n.s.	n.s.
Glu + Gln	0.15	0.96	n.s.	n.s.	n.s.	1.77	0.15	n.s.	n.s.	n.s.
GPC + PCh	0.33	0.86	n.s.	n.s.	n.s.	1.88	0.13	n.s.	n.s.	n.s.
GSH	0.74	0.57	n.s.	n.s.	n.s.	1.02	0.41	n.s.	n.s.	n.s.
Ins	1.73	0.16	n.s.	n.s.	n.s.	1.83	0.14	n.s.	n.s.	n.s.
MM09	0.25	0.91	n.s.	n.s.	n.s.	0.88	0.48	n.s.	n.s.	n.s.
MM09 + Lip09	0.45	0.77	n.s.	n.s.	n.s.	1.05	0.39	n.s.	n.s.	n.s.
NAA	4.67	**0.003**	0.5, 0.64	**4.4, 0.03 ***	1.9, 0.17	0.97	0.44	n.s.	n.s.	n.s.
NAA + NAAG	2.06	0.10	n.s.	n.s.	n.s.	1.03	0.41	n.s.	n.s.	n.s.
PCh	0.80	0.53	n.s.	n.s.	n.s.	2.60	**0.05**	2.3, 0.13	1.2, 0.34	1.1, 0.37
PCr	4.96	**0.002**	1.7, 0.21	**4.6, 0.02 ^+^**	2.1, 0.15	1.79	0.15	n.s.	n.s.	n.s.
Tau	1.23	0.31	n.s.	n.s.	n.s.	2.03	0.11	n.s.	n.s.	n.s.

BOLD = significant; Cr = creatine, PCr = phosphocreatine, GABA = *γ*-aminobutyric acid, Gln = glutamine, Glu = glutamate, GPC = glycero-phosphocholine, PCh = phosphocholine, GSH = glutathione, Ins = Inositol, MM = macromolecule, Lip = lipid, NAA = N-acteylasparate, NAAG = N-acetyl-aspartylglutamate, PCr = phosphocreatine, Tau = taurine; * 0.01, 50 μg R848 < 100 μg R848 at 3 h—0.04, 50 μg R848 < saline at 3 h; ^+^ 0.008, 50 μg R848 < 100 μg R848 at 3 h—0.04, 50 μg R848 < saline at 3 h. n.s. = no significant.

**Table 2 biology-11-00274-t002:** Statistical results for regions of interest delineated by the Allen Reference Atlas.

Region	Subregion	Size (mm^3^) *	3-Group × 3-Time MANOVAs		3-Group ANOVAs			*t*-Tests	
					Baseline	3 h	24 h		
			F_(4,42)_	*p*-Value	F_(2,24)_, *p*	F_(2,24)_, *p*	F_(2,24)_, *p*	*p*-Values	Comparisons
Isocortex (Cerebral cortex/Cortical plate)									
	Fronto-Orbital ^1^	5.2 ± 0.3	1.71	0.17	n.s.	n.s.	n.s.		
	Motor ^2^	19.5 ± 0.6	2.87	**0.04**	1.2, 0.32	**4.7, 0.02**	1.1, 0.36	**0.01**	**100** **μg R848 > 50** **μg R848 at 3 h**
	Somatosensory ^3^	25.4 ± 0.9	3.16	**0.02**	**3.4, 0.05**	**3.6, 0.05**	2.2, 0.14	**0.02**	50 μg R848 > 100 μg R848 at baseline
								**0.03**	**100** **μg R848 > 50** **μg R848 at 3 h**
								**0.03**	**100** **μg R848 > saline at 3 h**
	Insula ^4^	9.7 ± 0.5	4.43	**0.005**	**6.0, 0.009**	1.9, 0.18	0.6, 0.56	**0.003**	50 μg R848 > 100 μg R848 at baseline
								**0.03**	50 μg R848 > saline at baseline
	Temporal ^5^	6.9 ± 0.4	1.51	0.22	n.s.	n.s.	n.s.		
	Visual ^6^	21.6 ± 0.9	1.05	0.39	n.s.	n.s.	n.s.		
	Cingulate ^7^	6.6 ± 0.3	2.15	0.09	n.s.	n.s.	n.s.		
Olfactory Areas									
	Olfactory bulb	23.9 ± 1.2	1.77	0.15	n.s.	n.s.	n.s.		
	Olfactory cortex ^8^	22.7 ± 0.9	3.71	**0.01**	3.1, 0.07	**3.7, 0.04**	0.6, 0.57	**0.02**	**100** **μg R848 > saline at 3 h**
Hippocampal Formation ^9^		36.8 ± 1.4	1.33	0.28	n.s.	n.s.	n.s.		
Cortical Subplate		6.4 ± 0.3	0.80	0.53	n.s.	n.s.	n.s.		
Cerebral Nuclei									
	Striatum ^10^	35.1 ± 1.3	0.97	0.43	n.s.	n.s.	n.s.		
	Pallidum	8.4 ± 0.4	1.28	0.29	n.s.	n.s.	n.s.		
Brainstem									
	Thalamus ^11^	16.8 ± 0.6	0.73	0.57	n.s.	n.s.	n.s.		
	Hypothalamus	13.9 ± 0.5	2.18	0.09	n.s.	n.s.	n.s.		
Midbrain		35.2 ± 1.2	1.70	0.17	n.s.	n.s.	n.s.		
Hindbrain									
	Pons	17.1 ± 0.8	3.01	**0.03**	1.3, 0.29	**3.6, 0.04**	0.9, 0.43	**0.03**	**100** **μg R848 > 50** **μg R848 at 3 h**
								**0.03**	**100** **μg R848 > saline at 3 h**
	Medulla	36.5 ± 1.6	2.29	0.08	n.s.	n.s.	n.s.		
Cerebellar cortex		48.5 ± 1.9	1.99	0.11	n.s.	n.s.	n.s.		
CSF		10.7 ± 0.8	2.12	0.10	n.s.	n.s.	n.s.		
TOTAL ^⨍^		470.14 ± 15.0	2.40	0.07	n.s.	n.s.	n.s.		

* mean ± SD of 24 animals at baseline; ^⨍^ sum of 20 regions*1.15557 to account for exclusion of white matter by ATLAS; BOLD = significant; ^1^ frontal pole cerebral cortex + orbital area; ^2^ primary motor area + secondary motor area; ^3^ primary somatosensory area: trunk, lower limb, nose, upper limb, barrel field, mouth + supplemental somatosensory area; ^4^ gustatory areas + visceral area + agranular insular area; ^5^ auditory areas + temporal association areas; ^6^ visual areas + retrosplenial area + posterior parietal association areas; ^7^ anterior cingulate area + infralimbic area + prelimbic area; ^8^ accessory olfactory bulb + anterior olfactory nucleus + taenia tecta + dorsal peduncular area + piriform area + nucleus of the lateral olfactory + cortical amygdalar area + piriform-amygdalar area + post-piriform transition area; ^9^ ectorhinal area + perirhinal area + hippocampal region + retrohippocampal region; ^10^ striatum dorsal region + striatum ventral region + lateral septal complex + striatum-like amygdalar nuclei; ^11^ thalamus sensory-motor cortex related + thalamus polymodal association cortex related. n.s. = no significant.

## Data Availability

The data that support the findings of this study will be openly available at https://data.mendeley.com/ (uploaded on 26 January 2022).

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
