# Peer review of "Systemic Administration of the TLR7/8 Agonist Resiquimod (R848) to Mice Is Associated with Transient, In Vivo-Detectable Brain Swelling"

_biology, 2022, doi:10.3390/biology11020274_

Round 1
Reviewer 1 Report
The study by Zahr et al. presents interesting findings on in vivo brain responses to peripheral stimulation of the TLR7/8 receptors with resiquimod (R848), a modifier of immune response. The authors recorded body weight, temperature and brain volumes at baseline, 3 hours after drug injection, and 24 hours after the drug injection. Two different concentrations of R848 were used, 2 mg/kg and 4 mg/kg. They also used MR Spectroscopy to evaluate concentration of several metabolites in the dorsal striatum and hippocampus, including glutamine, N-acetyl aspartate, and phosphocreatine.
Sophisticated MRI segmentation approaches were used, and rigorous statistical methods were employed. In response to peripheral inflammation, the authors detected significant brain swelling 3 hours after R848 administration in the somatosensory cortex and pons. The manuscript advances the knowledge of brain responses to peripheral inflammation via TLR7/8 receptors and would be of interest to the readers of Biology. However, several weaknesses have to be addressed.
Minor points:
- Throughout the manuscript, “body weight” should be used instead of “weight”. The word ‘weight’ along can be misleading.
Figure quality needs to be improved:
- Figure 1: the title line in the legend is missing
- Figure 1: Arbitrary units (AU) should be used instead of Institutional units
- Figure 1: Overall, the legend is insufficient, and the abbreviations are not explained.
- Figure 2: the legend lacks the title line.
- Figure 2: Y axis: 5% should be indicated as 5, not 0.05
- Figure 2: “weight delta” should be rephrased
- Figure 6: again, e.g., 2% should be indicated as 2 on both axes, not 0.02
- Table 1 &2: The titles should be more elaborate, informative. Table grids should be removed.
- The term “olfactory cortex” is unusual, do the authors mean “piriform cortex”?
Author Response
Thank you for your time in reviewing this manuscript.
Your points were all appreciated and helped clarify and improve the manuscript.
Point by point responses follow.
- Throughout the manuscript, “body weight” should be used instead of “weight”. The word ‘weight’ along can be misleading.
Done
Figure quality needs to be improved:
- Figure 1: the title line in the legend is missing
Added.
- Figure 1: Arbitrary units (AU) should be used instead of Institutional units
Fixed.
- Figure 1: Overall, the legend is insufficient, and the abbreviations are not explained.
Fixed.
- Figure 2: the legend lacks the title line.
Added.
- Figure 2: Y axis: 5% should be indicated as 5, not 0.05
Fixed.
- Figure 2: “weight delta” should be rephrased
Changed to “Body weight at each scan relative to baseline scan body weights.”
- Figure 6: again, e.g., 2% should be indicated as 2 on both axes, not 0.02
Fixed.
- Table 1 &2: The titles should be more elaborate, informative. Table grids should be removed.
Grids removed and titles elaborated:
For Table 1, changed title from Statistical Results for Metabolites to >
Statistical results for MRS metabolites with average Cramér-Rao bounds <15% in hippocampal and striatal voxels
For table 2, changed title from Statistical Results for Regions of Interest to >
Statistical results for regions of interest delineated by the Allen Reference Atlas
- The term “olfactory cortex” is unusual, do the authors mean “piriform cortex”?
This has been clarified in Table 2 by adding a superscript and listing the regions that comprised the “olfactory cortex”.
Olfactory cortex = accessory olfactory bulb + anterior olfactory nucleus + taenia tecta + dorsal peduncular area + piriform area + nucleus of the lateral olfactory + cortical amygdalar area + piriform-amygdalar area + post-piriform transition area
Reviewer 2 Report
In this paper, the authors found an in vivo CNS response to R848. They also demonstrated that the brief metabolic changes at the lower dose may reflect energy imbalance and the transient volumetric changes (i.e., swelling) at the higher dose may represent brain edema in response to peripheral immune stimulation. They finally proposed the mechanism of pro‐inflammatory signal transduction from the periphery to the brain which may include stimulation of the vagus nerve; activation of cerebral endothelial cells; or direct access to brain parenchyma via circumventricular organs. Overall, this paper is well written with sufficient introduction, detailed methods and solid data. The topic is both interesting and important and will be helpful for future studies on the effects of immnue stimulation on the brain. However, the discussion part is relatively weak. The authors found significant volumetric changes in the cerebral cortex and the pons after the treatment, but not in the thalamus. In fact, the thalamus is a large structure and can be divided into multiple subregions. Subtle structral changes may be evident if the authors take a close look at the brain regions, the pulvinar nucleus in particular. Previous studies have shown that pulvinar is mutually and extensively connected with the prefrontal cortex, sensory cortex, superior colliculus and amygdala (Zhou et al., 2018) and plays very important roles in contextual multi-sensory processing and emotional response (Chou et al., 2020; Fang et al., 2020; Ibrahim et al., 2016). In addition, the dysfunction of pulvinar has been reported to be associated with abnormal immune response (Habek et al., 2009; Schweser et al., 2018). The authors should include the above key citations in the discussion. I would like to recommend this interesting paper to the editor after revision.
Zhou, Na et al. “The Mouse Pulvinar Nucleus Links the Lateral Extrastriate Cortex, Striatum, and Amygdala.” The Journal of neuroscience : the official journal of the Society for Neuroscience vol. 38,2 (2018): 347-362. doi:10.1523/JNEUROSCI.1279-17.2017
Chou, Xiao-Lin et al. “Contextual and cross-modality modulation of auditory cortical processing through pulvinar mediated suppression.” eLife vol. 9 e54157. 6 Mar. 2020, doi:10.7554/eLife.54157
Fang, Qi et al. “A Differential Circuit via Retino-Colliculo-Pulvinar Pathway Enhances Feature Selectivity in Visual Cortex through Surround Suppression.” Neuron vol. 105,2 (2020): 355-369.e6. doi:10.1016/j.neuron.2019.10.027
Habek, Mario et al. “Unusual cause of dementia in an immunocompetent host: toxoplasmic encephalitis.” Neurological sciences : official journal of the Italian Neurological Society and of the Italian Society of Clinical Neurophysiology vol. 30,1 (2009): 45-9. doi:10.1007/s10072-008-0007-5
Schweser, Ferdinand et al. “Mapping of thalamic magnetic susceptibility in multiple sclerosis indicates decreasing iron with disease duration: A proposed mechanistic relationship between inflammation and oligodendrocyte vitality.” NeuroImage vol. 167 (2018): 438-452. doi:10.1016/j.neuroimage.2017.10.063
Author Response
Thank you for your thoughtful comments regarding this manuscript. Please note our response to your request.
To our knowledge, no MRI studies have achieved the resolution necessary to permit volume quantification of the thalamic pulvinar nucleus in the rat. Indeed, successful thalamic segmentation in humans is still a work in progress as thalamic nuclei are largely invisible in conventional MRI due to poor contrast (1). Thus, while we appreciate that the thalamus is involved in the response to certain types of inflammation, it is beyond the scope of the current manuscript to describe all the brain regions potentially involved in the response to peripheral immune activation.
1. Datta R, Bacchus MK, Kumar D, Elliott MA, Rao A, Dolui S, et al. Fast automatic segmentation of thalamic nuclei from MP2RAGE acquisition at 7 Tesla. Magn Reson Med.